## Bacteriohopanepolyols track past environmental transitions in the Black Sea

Anna Cutmore<sup>1,2\*</sup>, Nora Richter<sup>1,3\*</sup>, Nicole Bale<sup>1</sup>, Stefan Schouten<sup>1,4</sup>, and Darci Rush<sup>1</sup>

2 3 4

5

6

7

1

\*Joint first authors

<sup>1</sup>NIOZ Royal Netherlands Institute for Sea Research, Texel, the Netherlands

<sup>2</sup>Durham University, Department of Geography, Durham, United Kingdom

<sup>3</sup>EAWAG Swiss Federal Institute of Aquatic Science and Technology, Dübendorf, Switzerland

<sup>4</sup>Utrecht University, Utrecht, the Netherlands

8 9 10

Correspondence to: Anna Cutmore, anna.v.cutmore@durham.ac.uk and Nora Richter, nora.richter@eawag.ch

1112

#### **Abstract**

Bacteriohopanepolyols (BHPs) are structurally diverse compounds produced by a wide range of bacteria making them ideal candidates as chemotaxonomic biomarkers and indicators of bacterially-driven biogeochemical processes in the geological record. In this study, we characterize changes in the BHP distribution in the Black Sea over the past 20 thousand years (ka), as the basin underwent three distinct environmental phases: (i) an oxic lacustrine phase where the Black Sea was disconnected from the global ocean; (ii) a transition period marked by the initial influx of marine water into the basin; and (iii) a marine phase where the basin was permanently euxinic. During the lacustrine phase we observe a high abundance and diversity of nucleoside BHPs (Nu-BHPs) that are likely derived from elevated terrigenous inputs as well as production of Nu-BHPs in the brackish-to-fresh water column. The transition phase is marked by a decrease in the abundance of most Nu-BHPs and an increase in the abundance of methoxylated-BHPs as well as BHPs such as aminobacteriohopanetriol which are ubiquitous across a wide range of environments including soils as well as marine and freshwater settings. The euxinic marine phase (7.2 ka-present) can be divided into two stages based on changes in BHP composition. The early stage is characterised by a high abundance of aminobacteriohopanetetrol and aminobacteriohopanepentol, which were likely produced by methanotrophs at the oxycline. A shallow oxycline likely allowed for increased transport of these BHPs to the sediment. The later marine phase is characterised by a decline in these BHPs, likely due to a deepening of the oxycline and reduced transport of BHPs from the oxycline to the sediment. The changes in BHP distributions throughout the record may either be attributed to shifts in the bacterial communities or physiological adaptations of bacteria to the changing environment. Throughout the record, diagenetic products of BHPs (e.g., anhydro-bacteriohopanetetrol) were detected. These degradation products, however, remain a small proportion of the overall BHP composition, indicating excellent preservation conditions throughout the record. This study offers new insights into changes in microbial communities and biogeochemical processes that occurred in the Black Sea during the Last Deglaciation and Holocene in response to substantial shifts in the hydrology and oxygen conditions of the basin.

343536

Keywords: Bacteriohopanepolyols, Black Sea, Holocene, Last Deglaciation, Anoxia, Euxinia

#### 1. Introduction

Bacteriohopanepolyols (BHPs) are pentacyclic triterpenoids produced by a wide-range of bacteria that are considered to be the precursors to one of the most abundant lipids in the geological record, hopanoids (Rohmer et al., 1984; Ourisson and Albrecht, 1992). BHPs themselves are also abundant in the geological record and are wellpreserved in sedimentary archives for >1 million years (Handley et al., 2010). In some cases, BHPs are linked to certain source organisms (Cvejic et al., 2000; Kool et al., 2014; Rush et al., 2014; van Winden et al., 2012) or biogeochemical processes (Talbot et al., 2003; 2007; Coolen et al., 2008; Blumenberg et al., 2013). Consequently, shifts in BHP compositions in the geological record have become a useful tool for palaeoclimate reconstructions to assess changes in redox conditions (Blumenberg et al., 2013; Matys et al., 2017; Rush et al., 2019; Zindorf et al., 2020), salinity (Coolen et al., 2008), methanotrophy (Talbot et al., 2014), and soil inputs (Blumenberg et al., 2013). For example, 35-aminobacteriohopane-31,32,33,34-tetrol (aminotetrol from herein), 35-aminobacteriohopane-30,31,32,33,34-pentol (aminopentol from herein), and methylcarbamate-amino BHPs are primarily associated with aerobic methane-oxidizing bacteria (MOB; Rohmer et al., 1984; Jahnke et al., 1999; Cvejic et al., 2000; Talbot et al., 2001; Zhu et al., 2011; van Winden et al., 2012; Rush et al., 2016) and have been used to reconstruct past changes in methane-oxidation in the Congo River Delta (Talbot et al., 2014; Spencer-Jones et al., 2017), the Baltic Sea (Blumenberg et al., 2013), and in an Antarctic lake (Coolen et al., 2008). Recent studies show that bacteriohopane-32,33,34,35-tetrol-x (BHT-x) is synthesized by anaerobic ammonium oxidizing (anammox) bacteria (Rush et al., 2014; Schwartz-Narbonne et al., 2020) and thus has been used to reconstruct changes in N-cycling under low-oxygen conditions in Mediterranean sapropels (Rush et al., 2019; van Kemenade et al., 2023) and the Black Sea (Cutmore et al., 2025). Nucleoside BHPs (Nu-BHPs) occur in high abundance in soils (Seemann et al., 1999; Bravo et al., 2001; Cooke et al. 2008a; Xu et al., 2009; Rethemeyer et al., 2010), and, therefore, have been applied to track the transport of soil organic matter to aquatic sediments (Talbot and Farrimond, 2007; Cooke et al., 2008b). Recent studies, however, have identified potential production of Nu-BHPs along redoxclines in marine oxygen minimum zones (Kusch et al., 2021) and near the oxycline in lakes (Richter et al. 2023). While this highlights a small number of the known BHPs and their potential application to paleo-records, there are still many unknowns surrounding the origin and sources of other BHPs in the environment. Recent analytical advancements in BHP analysis has led to the identification of additional novel BHPs in both cultures and environmental samples (Talbot et al., 2016a; Hopmans et al., 2021; Richter et al., 2023). The large structural diversity of BHPs along with their preservation potential in the recent geological record, points to a useful new tool in biomarker research that can provide insights into past changes in microbial communities and/or biogeochemical processes.

The Black Sea provides a compelling location to study the impact of bacterial community composition and environmental changes on BHP composition due to its unique hydrological history and present-day characteristics. Today, it is the largest permanently stratified anoxic basin in the world with limited connection to the global ocean through the Bosporus Strait. Over the past 20 thousand years (ka), however, the Black Sea underwent significant hydrological changes (Fig. 1). During the Last Glacial Maximum (LGM), the basin was an oxygenated brackish-to-

freshwater environment (Schrader, 1979), then, during the subsequent deglaciation, the basin experienced many environmental shifts, including changes in temperature (Bahr et al., 2005; 2008; Ion et al., 2022), water-level (Ivanova et al., 2007; Nicholas et al., 2011; Piper and Calvert, 2011), and freshwater input. This freshwater influx was influenced by changes in regional precipitation and melting of Eurasian ice sheets and alpine glaciers (Bahr et al., 2005; 2006; 2008; Badertscher et al., 2011; Shumilovskikh et al., 2012). The basin reconnected with the global ocean around 9.6 ka when post-glacial sea-level rise led to an initial marine inflow (IMI) over the Bosporus sill (Aksu et al., 2002; Major et al., 2006; Bahr et al., 2008; Ankindinova et al., 2019), leading to higher water column salinity (Marret et al., 2009; Verleye et al., 2009; Filipova-Marinova et al., 2013) and permanent euxinia developing in the water column after ~7.2 ka (Arthur and Dean, 1998; Eckert et al., 2013).

Several studies have demonstrated the usefulness of lipid biomarkers as records of biogeochemical processes and as indicators of ecological niches within the modern-day water column and surface sediments of the Black Sea (Schubert et al., 2006; Coolen et al., 2007; Wakeham et al., 2007; Schubotz et al., 2009; Sollai et al., 2019; Bale et al., 2021; Kusch et al., 2022). BHPs produced in the lower suboxic zone and upper sulfidic zone, for instance, were shown to record activities of aerobic methanotrophs and anammox bacteria, respectively, demonstrating their reliability as potential biomarkers for microbial communities (Kusch et al., 2022). Furthermore, previous work has explored BHP changes in the Black Sea over the Holocene, however, this applies to a limited number of BHPs (Blumenberg et al., 2009a). With significant advances in analytical techniques, there is now the opportunity to explore the BHP geolipidome of the Black Sea in unprecedented detail. Indeed, we recently used the ratio of bacteriohopanetetrol (BHT)-34S and the later eluting stereoisomer BHT-x to trace past anammox activity in the Black Sea over the last 20 ka, complementing other nitrogen cycling biomarker tools (Cutmore et al., 2025). In this study, we now fully characterize the distribution of BHPs in a Black Sea sediment core over the Last Deglaciation and Holocene to shed light on changing environmental conditions and bacterial communities.

### 2. Methods

Piston core 64PE418 (235 cm length) was recovered from the Black Sea (42°56 N, 30°02 E; 1970 metres below sea level [mbsl]) during the 2017 Pelagia cruise (Fig. 1). The core was subsampled at the Royal Netherlands Institute for Sea Research, and all samples were stored frozen until further analysis. The contemporary Black Sea water column is characterized by pronounced redox gradients. At the location of the core site in the western gyre of the basin, the oxic zone, found between 0-75 m depth range, has an oxygen concentration of ~121  $\mu$ mol/kg at 50 m depth, while sulphide concentrations are not detected (Sollai et al., 2019). The suboxic zone lies below, situated between 75-115 m depth range, with traces of sulphide found at the bottom of this layer between 105-115 m (Sollai et al., 2019). Beneath this is the euxinic zone, located between 115-2000 m depth range, with sulphide concentrations increasing significantly with depth, reaching ~400  $\mu$ mol/L at 2,000 m water depth (Sollai et al., 2019).

### 2.1 Lipid extraction and analysis

Sediment samples (n=44) were taken at 5 cm intervals throughout the core's depth before freeze drying. Using a modified Bligh and Dyer extraction method (as described previously by Bale et al., 2021), lipids were extracted from the dry sediment samples, as described in detail in Cutmore et al. (2025). Extraction blanks were carried out alongside the sediment extractions, using the same glassware, solvents and extraction methodology.

Extract analysis was performed on an ultra-high performance liquid chromatography-high resolution mass spectrometer (UHPLC-HRMS) in reverse phase as described in Hopmans et al. (2021). An Agilent 1290 Infinity I UHPLC, featuring a thermostatted auto-injector and column oven, was utilised, coupled to a Q Exactive Orbitrap MS with Ion Max source with heated electrospray ionization (HESI) probe (Thermo Fisher Scientific, Waltham, MA). Separation was carried out with an Acquity BEH C18 column (Waters, 2.1 × 150 mm, 1.7 μm) maintained at 30°C, using an eluent composition of (1) MeOH/H<sub>2</sub>O/formic acid/14.8 M NH<sub>3</sub>ag [85:15:0.12:0.04 (v:v)] and (2) IPA/MeOH/formic acid/14.8 M NH₃aq [50:50:0.12:0.04 (v:v)]. The flow rate was 0.2 mL min<sup>-1</sup> with an elution program of: 95% A for 3 minutes, followed by a 40% linear gradient at 12 minutes, then 0% A at 50 minutes which was maintained until 80 minutes. The positive ion HESI parameters were as follows: capillary temperature at 300°C; sheath gas (N<sub>2</sub>) pressure at 40 arbitrary units (AU); spray voltage at 4.5 kV; auxiliary gas (N<sub>2</sub>) pressure at 10 AU; Slens at 70 V; probe heater temperature at 50°C. Lipid analysis was conducted across a mass range of m/z 350–2000 (resolving power 70,000 ppm at m/z 200), followed by data-dependent tandem MS/MS (resolving power 17,500 ppm), in which the 10 most abundant masses in the mass spectrum were fragmented successively. Optimal fragmentation was achieved with a stepped normalized collision energy of 22.5 and 40 (isolation width 1.0 m/z) for BHP analysis. The Q Exactive was calibrated to a mass accuracy of ±1 ppm using the Thermo Scientific Pierce LTQ Velos ESI Positive Ion Calibration Solution. During analysis, dynamic exclusion was applied to temporarily exclude masses for 6 s, enabling selection of less abundant ions for MS/MS.

BHPs were tentatively identified based on their retention time, exact mass, and fragmentation spectra as described in Hopmans et al. (2021) and Richter et al. (2023). Note, the BHPs discussed in this study are tentative structures and we rely on comparisons with previous studies for structural elucidation, in particular for determining the stereochemistry of the bacteriohopanetetrol isomers (Schwartz-Narbonne et al., 2020). Integrations were performed on (summed) mass chromatograms of relevant molecular ions ([M+H]+, [M+NH4]+, and [M+Na]+). The BHP absolute abundances are all presented as peak area per gram of total organic carbon (TOC) since we lack appropriate standards to correct for differences in ionization efficiency between BHPs (Cutmore et al., 2025).

#### 2.2. Age Model

The age model for core 64PE418 was previously published in Cutmore et al. (2025). Seven bulk organic matter <sup>14</sup>C dates were used in its production. Six of these were from core 64PE418, combined with an additional bulk organic carbon <sup>14</sup>C date from core KNR 134-08 BC17 (Jones and Gagnon, 1994), which is sourced from the same location and water depth as 64PE418. This additional <sup>14</sup>C date is from the widely observed Unit I/II boundary and was employed to refine the age model for the upper section of the core. Variable reservoir-ages, calculated by

Kwiecien et al. (2008) for intermediate water depths in the Black Sea over the last ~20 ka, were added to the calibration. The  $^{14}$ C dates were calibrated using the Marine20 calibration curve (Heaton et al., 2020) for the upper three  $^{14}$ C dates (reflecting the period after the IMI), while the lower four  $^{14}$ C dates were calibrated using the IntCal20 calibration curve (Reimer et al., 2020), (reflecting the period when then Black Sea was a lacustrine environment). The R-code, CLAM (Blaauw, 2010), was used to create the age—depth model. The BHP record from core 64PE418 spans the last 19.5 ka, with an average resolution of ~450 years. The key transitions (identified by colour and elemental changes in the core [Cutmore et al., 2025]) are as follows: the onset of the IMI is dated at 9.6 ka  $\pm$  237 yrs; Unit II/III occurs at 7.2 ka  $\pm$  202 yrs; and the Unit I/II boundary is dated at 2.6 ka  $\pm$  402 yrs.

#### 2.3. Statistical analyses

A principal component analysis (PCA) was used to assess the variability in BHP distributions downcore after applying a Hellinger transformation and scaling the BHP dataset. All analyses were performed and visualized in R (version 4.2.2; R Core Team, 2023) with the vegan package (version 2.6-4; Oksanen et al., 2020), factoextra (version 1.0.7; Kassambara and Mundt, 2020), FactoMineR (version 2.7; Lê et al., 2008), and ggplot2 (version 3.4.0; Wickham and Chang, 2016).

#### 3. Results and Discussion

Significant changes in total organic carbon (TOC) content, colour and elemental signatures of the core are observed (Cutmore et al., 2025), corresponding to changing environmental conditions in the Black Sea basin over the last 20 ka (Arthur and Dean, 1998; Bahr et al., 2005; Jones and Gagnon, 1994; Ankindinova et al., 2019). Based on this, the core is divided into three widely acknowledged key periods: the oxic lacustrine phase (19.5-9.6 ka) where the basin was disconnected from the global ocean; the transition phase (9.6 - 7.2 ka) after the IMI over the Bosporus sill at ~9.6 ka, whereby the basin moved towards a marine environment; and the marine phase (7.2 ka to the present) where the Black Sea became a euxinic brackish-to-marine environment (Jones and Gagnon, 1994; Arthur and Dean, 1998; Aksu et al., 2002; Major et al., 2006; Bahr et al., 2008; Ankindinova et al., 2019; Cutmore et al., 2025).

#### 3.1 BHP distribution changes over time

In total, 63 BHPs were detected throughout the Black Sea record. BHP structures and other compounds discussed in this study are shown in Fig. 2. A PCA of the relative distribution of BHPs reveals four distinct clusters (Fig. 3), which correspond with the above defined phases of the Black Sea: an oxic lacustrine phase (19.5 - 9.6 ka), a transition phase (9.6 - 7.2 ka), and a marine phase that is further divided into an early marine phase (7.2 - 5 ka) and a late marine phase (5 ka to present). PC1 explains 40.8% of the variance and is positively correlated with Nu-BHPs, while PC2 explains 18.2% of the variance and is negatively correlated with BHPs such as 35-aminobacteriohopane-32,33,34-triol (aminotriol from herein), N-formylated-aminotriol, and ethenolamine-BHT are present throughout the

record (Fig. 4), which is in line with the ubiquitous nature of these BHPs that are found in contemporary marine, freshwater and terrestrial environments (Talbot and Farrimond 2007; Hopmans et al., 2021; Richter et al., 2023). Note, due to a lack of standards, absolute concentrations cannot be determined and BHP relative abundances must be interpreted with caution. However, despite this limitation, this semi-quantitative approach still provides a valuable overview of major changes in BHP distributions throughout the record.

Each phase of the Black Sea is defined by several notable changes in BHP distributions. Throughout the oxic lacustrine phase, during the Last Deglaciation and early Holocene (19.5 – 9.6 ka), a wider range of BHPs were detected compared to other phases (Fig. 4). This coincides with low TOC levels and low concentrations of elements that accumulate in sediments under anoxic conditions (i.e., uranium [U], vanadium [V], and molybdenum [Mo]) (Cutmore et al., 2025), pointing to a well-oxygenated environment. Nu-BHPs, for instance, are present in higher absolute abundance and demonstrate a more diverse distribution during the lacustrine phase relative to the other periods (Fig. 3). Bacteriohopane-32,33,34-triol (BHtriol), bacteriohopane-31,32,33,34,35-pentol (BHpentol), and bacteriohopane-30,31,32,33,34,35-hexol (BHhexol) are also present in higher absolute abundance during the oxic lacustrine phase, relative to the rest of the record where they are largely absent. The transition phase (9.6 - 7.2 ka) occurs after the IMI at ~9.6 ka and is characterized by a decrease in almost all Nu-BHPs with the exception of an increase in aminotriol, ethenolamine-BHT, and N-formylated-aminotriol.

At 7.2 ka, there is a shift to a permanent euxinic marine system which led to increased preservation of organic matter resulting from permanent water column anoxia. Despite the increased preservation, previous studies have reported a decrease in both homohopanoid and BHP concentrations in the Black Sea during this period (Blumenberg et al., 2009a) indicating either a decline in BHP production or a decrease in BHP transport to the sediment, as observed in the modern-day Black Sea water column (Wakeham et al., 2007; Blumenberg et al., 2009a; Kusch et al., 2022). In our record, the early marine phase (7.2 – 5 ka) is marked by an increase in methoxylated-BHT (methoxy-BHT from herein), methoxylated-ethenolamine-BHT (methoxy-ethenolamine-BHT from herein), and propenolamine-BHT. There is a temporary increase in aminotetrol, aminopentol, N-acylated-aminotriols, ethenolamine-BHpentol, and ethenolamine-BHhexol that spans the transition of the early and late marine phases. The absolute abundances of individual BHPs are considerably lower after ~3.9 ka, with the exception of methoxy-BHT and methoxy-ethenolamine-BHT, which increase from 0.9 ka to present (Fig. S2-S7). A decrease in BHP and homohopanoid concentrations during the late marine phase has also been reported in a previous reconstruction of the Black Sea (Blumenberg et al., 2009a). This highlights how distinct BHP distributions characterize the different environmental phases of the Black Sea record.

#### 3.2. Diagenetic products of BHPs

BHP preservation was assessed throughout the core by tracking known early diagenetic products of BHP degradation: 32,35-anhydrobacteriohopanetetrol (anhydro-BHT) and anhydrobacteriohopanepentol (anhydro-BHpentol) which are formed from the dehydration and cyclization of BHT and BHpentol, respectively (Bednarcyzk et

al., 2005; Talbot et al., 2005; Schaeffer et al., 2008; 2010). In addition, it has been shown that anhydro-BHT is also a degradation product of composite BHPs, such as BHT-cyclitol ether and adenosylhopane (Schaeffer et al., 2010; Eickhoff et al., 2014). Note, the diagenetic BHPs discussed here are only the early degradation products of BHPs, and we do not discuss BHPs preserved as e.g. organic sulfur compounds (a novel C<sub>35</sub> hopanoid in sediments [Valisolalao et al., 1984]) or the fate of C<sub>27</sub>-C<sub>35</sub> hopanes. However, previous studies have shown that the concentrations of S-bound hopanoids only make up to 7.5% of the sum of free BHPs and S-bound hopanoids, with the lowest abundance of S-bound hopanoids occurring during the lacustrine phase (Blumenberg et al. 2009a). Non-extended hopanoids were found to be more abundant during the marine phase and decreased with depth, further non-extended hopanoids were found to be less abundant than BHPs during the transition and lacustrine phases of the Black Sea record (Blumeberg et al., 2009a).

Throughout the record, the absolute abundance of anhydro-BHT and -BHpentol follow opposite trends to each other, with a higher absolute abundance of anhydro-BHpentol and anhdyro-BHT during the lacustrine oxic phase and marine phase, respectively (Fig. S2 and Fig. S8). In a previous study of Black Sea surface sediments, anhydro-BHT increased below the top 2 cm of the core, suggesting rapid early diagenetic modifications of BHPs (Kusch et al., 2022). Previous studies point to the possibility of selective degradation of BHPs (Kusch et al., 2022), in particular the degradation of Nu-BHPs relative to co-occurring composite BHPs, such as aminotriol (Talbot et al., 2016b). However, Nu-BHPs occur in high absolute abundance throughout the record, particularly during the lacustrine oxic phase where we would anticipate increased degradation due to an oxygenated water column relative to the more recent marine phase whereby the water column became anoxic (Schrader, 1979; Arthur and Dean, 1998; Eckert et al., 2013).

BHtriol and unsaturated BHtriol, found previously in lacustrine environments (Rodier et al., 1999; Watson and Farrimond, 2000), follow a similar trend to anhydro-BHpentol, with all three BHPs present in higher absolute abundance during the lacustrine phase relative to the late transition period and the marine phase, suggesting that BHtriol and unsaturated BHtriol could also be diagenetic products (Fig. S2 and Fig. S8). Indeed, BHtriol only contains 7 carbons on the side-chain instead of the typical 8 carbons found in other BHPs and in unsaturated BHtriol the unsaturation occurs on the side-chain, indicating that they may have formed from decarboxylation and dehydration of functionalized BHPs. It is possible that other early diagenetic products of BHPs (i.e. diols and triols identified in Watson & Farrimond, 2000) were present during the lake phase, however, the mass spectra were not clear enough to identify these compounds.

The decline in BHtriol and anhydro-BHpentol throughout the transition period (between 9.6 - 7.2 ka) may indicate reduced degradation due to intermittent anoxia in the water column. It may, however, also indicate a decrease in the abundance of composite BHPs that are precursors to these diagenetic BHPs, with both BHhexol and BHpentol also declining in absolute abundance during the transition period compared to the lacustrine phase. Anhydro-BHT, in contrast, does not decline during the transition period, instead increasing during the transition and marine periods. This could suggest that there is an increase in the source of the diagenetic precursors for anhydro-

BHT (i.e., BHT and adenosylhopane; Eickhoff et al., 2014), while the precursors to the other diagenetic BHPs decreased. The changes in the absolute abundance of these BHPs are likely associated with the changes in precursor BHP concentrations. Overall, there are BHP diagenetic products present throughout the record, averaging 11% during the lacustrine period, 6% during the transition, and 11% and 16% during the early and late marine period, respectively. These results taken together with previous analyses of S-bound hopanes and non-extended hopanoids (Blumenberg et al., 2009a), suggests generally good preservation conditions throughout the record.

#### 3.3. Sources of Nu-BHPs

Nu-BHPs were particularly abundant and diverse during the lacustrine phase (Fig. 5), with the highest Nu-BHP abundance occurring between 19.5 - 13.8 ka, during the last deglaciation, dominated by adenosylhopane<sub>HG-dlMe</sub> followed by adenosylhopane and adenosylhopane<sub>HG-Me</sub>. Many of the Nu-BHPs are present in higher abundance early in the record and decline during the later part of the lacustrine phase. This pattern corresponds with changes in terrigenous input, as indicated by a decline in the elemental records of titanium to calcium (Ti/Ca) and of potassium (K) (Cutmore et al., 2025). Furthermore, it corresponds with a decrease in the branched and isoprenoid tetraether (BIT) index over the lacustrine period (Bingjie Yang, personal communications, 2024), which indicates a decline in the delivery of terrestrial organic matter to our site (Hopmans et al., 2004), likely primarily supplied by the Danube, river. Between 18 and 14.8 ka, there were multiple significant meltwater pulses in the Black Sea (Major et al., 2006; Bahr et al., 2008; Badertscher et al., 2011; Yanchilina et al., 2019), followed by higher precipitation during the Bølling Allrød (BA) (14.5-13 ka [Shumilovskikh et al., 2012]). Consequently, the increased terrestrial input is likely the result of this increased freshwater influx. Many of the Nu-BHPs may therefore be sourced from this increased contribution of terrestrial organic matter, with Nu-BHPs primarily associated with soils (Talbot and Farrimond, 2007; Cooke et al., 2008a; Xu et al., 2009; Rethemeyer et al., 2010).

The absolute abundance of Nu-BHPs declines after 13 ka, corresponding with low Ti/Ca and K values (Cutmore et al., 2025), which indicate reduced input of terrestrial organic matter. This corresponds with drier regional conditions during the Younger Dryas (YD) and Early Holocene (Shumilovskikh et al., 2012) and reduced influx of meltwater into the Black Sea due to the retreat of ice sheets across Northern Europe (Major et al., 2006). During this time, the composition of the Nu-BHPs also changes with a decline in the absolute abundance of adenosylhopane<sub>HG-diMe</sub>. After 13 ka, 2Me-adenosylhopane<sub>HG-diMe</sub> increases, peaking at 10.6 ka and declining abruptly at the IMI at 9.6 ka. While the majority of Nu-BHPs identified in this record are likely linked to soil input and closely follow the terrigenous elements, some, such as 2Me-adenosylhopane<sub>HG-diMe</sub>, demonstrate a different pattern at 11 ka. 2Me-adenosylhopane<sub>HG-diMe</sub>, for instance, peaks at the start of the Holocene (11-9 ka) followed by an abrupt decline at the IMI when anoxic conditions started to prevail in the lower part of the water column (Fig. S7). 2Me-adenosylhopane<sub>HG-diMe</sub> was previously detected in higher abundance near the chemocline of a lake (Richter et al., 2023), suggesting potential in situ production rather than a terrestrial source for this Nu-BHP between 11-9 ka.

During the transition period, most Nu-BHPs are still detected in the record (e.g., adenosylhopane<sub>HG-Me</sub> and 2Me-adenosylhopane<sub>HG-diMe</sub>), but in much lower absolute abundance than during the oxic lacustrine period, likely due to reduced input of terrestrial organic matter, corresponding with low Ti/Ca and K values (Cutmore et al., 2025). Me-adenosylhopane<sub>HG-diMe</sub>, however, demonstrates a higher abundance during the transition period compared to the end of the oxic lacustrine period (Fig. S6). As this compound has previously been found near the chemocline in lakes (Richter et al., 2023), it is possible that it was being produced near the oxic-anoxic transition zone during periods of water column anoxia.

The diversity of Nu-BHPs is low throughout the marine phase with many Nu-BHPs below detection limit throughout this period (e.g., 2Me-Adenosylhopane, 3Me-Adenosylhopane, diMe-adenosylhopane<sub>HG-Me</sub>, 3Me-Adenosylhopane<sub>HG-diMe</sub>, 2Me-N1-methyl-inosylhopane and Me-N1-methyl-inosylhopane). The Nu-BHPs that are detected are present in low absolute abundance compared to other phases. This is likely due to the low contribution of terrestrial organic matter to the site, demonstrated by the low Ti/Ca values (Cutmore et al., 2025), which reduced the delivery of soil-sourced Nu-BHPs to this site. Notably, there is an increase in several Nu-BHPs after 1 ka, including adenosylhopane, 2Me-adenosylhopane<sub>HG-Me</sub>, Me-adenosylhopane<sub>HG-diMe</sub>, inosylhopane, N1-methylinosylhopane and Me-N1-methyl-inosylhopane. If these BHPs are not derived from soil inputs, then they are likely being produced in the water column of the Black Sea. Indeed, adenosylhopane, 2Me-adenosylhopane<sub>HG-Me</sub>, and inosylhopane are detected in the modern-day Black Sea water column (Kusch et al., 2022).

In summary, most Nu-BHPs detected in the Black Sea core are likely sourced from soils, linked to freshwater influx into the basin. Several Nu-BHPs, such as adenosylhopane, 2Me-adenosylhopane<sub>HG-Me</sub>, Me-adenosylhopane<sub>HG-diMe</sub>, consylhopane, N1-methylinosylhopane and Me-N1-methyl-inosylhopane, however, exhibit patterns that suggest these Nu-BHPs were produced in the water column of the Black Sea. As ongoing research continues to explore the sources of Nu-BHPs both in soils and in the water column of lakes and marine environments, it may be possible to expand the interpretation of these records.

#### 3.4. BHPs associated with the N-cycle

BHT-x, produced by anaerobic ammonium oxidising (anammox) bacteria, was previously reported, revealing increased anammox activity after 7.2 ka leading to the loss of bioavailable nitrogen during the marine euxinic period (Cutmore et al., 2025). To further advance our understanding of the nitrogen cycle over the last 20 kyr in the Black Sea, we highlight other BHPs as potential biomarkers for investigating this biogeochemical cycle.

Oxazinone-aminotriol, for instance, was detected during the lacustrine and transition phases, but absent during the marine euxinic period. As a potential marker for nitrite-oxidizing bacteria (NOB) (Elling et al., 2022; Richter et al., 2023), its presence during the lacustrine and transition phases suggests that nitrite oxidation, performed by NOB as the second step of nitrification (Stein & Nicol, 2018), was an important process when the upper water column was oxygenated. Its absence during the marine phase indicates a decline under euxinic conditions. This pattern mirrors the crenarchaeol record from the same core (Cutmore et al., 2025), a biomarker exclusively produced by the

abundant and widespread archaea Thaumarchaeota (Nitrososphaerota) (Sinninghe Damste et al., 2002), which performs the crucial first step in nitrification in the Black Sea (Lam et al., 2007) by aerobically oxidizing ammonia to nitrite (Könneke et al., 2005; Wuchter et al., 2006). This reinforces the interpretation that nitrification was enhanced when the water column was well-oxygenated.

Other BHPs identified in this study may also be associated with the nitrogen-cycle. Notably, BHT-CE abundance closely tracks that of crenarchaeol (Fig. 6; Cutmore et al., 2025), indicating a potential association between the dominant bacterial producer of BHT-CE and the ammonia oxidizing archaea, Nitrososphaerota. This pattern may reflect a functional or ecological coupling between the dominant BHT-CE-producing bacteria and the nitrification process in the Black Sea. Similar associations between archaea and bacteria have been observed elsewhere, such as between ca. *Methanoperedens nitroreducens* and ca. *Methylomirabilis oxyfera* (NC10 phylum) during methane oxidation (Smit et al., 2019), and with anammox bacteria utilising nitrite produced by Nitrososphaerota and ammonia-oxidising bacteria (AOB) in the modern-day Black Sea (Kuypers et al., 2003; Lam et al., 2007). Although BHT-CE has been detected in an enrichment culture from a peat sample for the NC10 bacteria, its exact microbial source remains uncertain (Kool et al., 2014). Consequently, a functional coupling between Nitrososphaerota and an unknown BHT-CE-producing bacteria may have existed in the Black Sea water column. Alternatively, both microbes may have inhabited similar ecological niches and depth habitats, leading to parallel responses to changes in water column conditions.

We identified BHPs with a methylation in the second carbon position of the A ring (2Me-BHPs) in our record. 2Me-BHT was largely absent during the lacustrine phase until the early Holocene (10.3 ka), where it increased before peaking at the start of the transition phase (9.6 ka) and again increasing during the marine phase (6-4.2 ka) (Fig. S2). 2Me-BHPs, precursors for 2Me-hopanes, were initially linked to cyanobacteria (Summons et al., 1999; Kuypers et al., 2004; Talbot et al., 2008), but are now known to also be produced in low amounts by alpha ( $\alpha$ ) -proteobacteria (Rashby et al., 2007; Welander et al., 2010; Naafs et al., 2022), such as NOB *Nitrobacter vulgaris* (Elling et al., 2020), as well as Actinobacteria and one strain of Acidobacteria (Welander et al., 2010; Sinninghe Damsté et al., 2017; Naafs et al., 2022). In our record, the increase in 2Me-BHT coincides with increasing absolute abundance of hexose HGs, known indicators of increased N<sub>2</sub>-fixation by freshwater or brackish heterocystous cyanobacteria (Cutmore et al., 2025). It is therefore possible that 2Me-BHT is related to N<sub>2</sub>-fixing cyanobacteria in the Black Sea record, with 2Me-BHT previously identified in various heterocystous cyanobacteria, including *Nostoc* sp., *Calothrix* sp. and *Chlorogloeopsis fritschii* (Talbot et al., 2008). While  $\alpha$ -proteobacteria may also contribute to the presence of 2Me-BHT in this record, the lack of alignment with oxazinone-aminotriol (a potential marker for NOB [Elling et al., 2022]) supports the theory that the primary origin of 2Me-BHT in this record is from N<sub>2</sub>-fixing cyanobacteria.

While further research is needed to refine their use, the BHPs in this record show potential as both novel indicators for microbial nitrogen cycling and as supporting evidence for established biomarkers like those of heterocystous cyanobacteria.

#### 3.5. BHPs associated with methane-oxidizing bacteria

Aminotetrol and aminopentol are present throughout the lacustrine phase of the Black Sea, albeit in low absolute abundance compared to the other phases (Fig. S3). Aminotetrol and aminopentol are associated with MOB (Rohmer et al., 1984; Neunlist and Rohmer, 1985a,b; Jahnke et al., 1999; Cvejic et al., 2000; Talbot et al., 2001; Zhu et al., 2011; van Winden et al., 2012; Rush et al., 2016), and are produced in small proportions by sulfate-reducing bacteria (SRB) (Blumenberg et al., 2006; 2009b; 2012). Given the low sulfate availability and aerobic conditions during the lacustrine phase, (Cutmore et al., 2025), aminotetrol and aminopentol were more likely being produced by methane-oxidizing bacteria than SRB, which are considered obligate anaerobes (Hao et al., 1996). Although NOB have also been proposed as a source of aminopentol and aminotetrol (Elling et al., 2022), the lack of co-variation with other biomarkers associated with the nitrogen-cycle (i.e. crenarchaeol), suggest that NOB are not the main contributors of these BHPs. Other independent biomarkers, such as high concentrations of  $17\beta(H)$ -moret-22(29)ene with depleted  $\delta^{13}$ C values, support a possible aerobic methanotrophic source (Uemura and Ishiwatari, 1995; Blumenberg et al., 2009a). The continuous presence of dioxanone-methylaminotriol, which was previously detected in surface sediments and bottom waters of lakes that experience seasonal hypoxia or anoxia (Richter et al., 2023), throughout this period suggests that the lake phase of Black Sea may have experienced seasonal stratification and the formation of an oxycline. In modern lakes, methane-oxidizing bacteria are known to thrive at the oxycline of the water column (Hanson & Hanson, 1996), but are also active at the sediment-water interface in well-mixed lakes (e.g., Kuivila et al., 1988; Frenzel et al., 1990), in anoxic parts of the water column (Schubert et al., 2010; Blees et al., 2014; Oswald et al., 2016), and in anoxic sediments (Martinez-Cruz et al., 2017; He et al., 2021; Su et al., 2022). Although it is unclear how much of these different pools are contributing to the sedimentary aminotetrol and aminopentol pool identified in the Black Sea record, we can still conclude that aminotetrol and aminopentol are likely sourced from methane-oxidizing bacteria during the lacustrine phase.

Several BHPs increased during the transition period relative to the preceding lacustrine phase (i.e., methoxy-BHT, aminotriol, aminotetrol, EC-aminotriol, ethenolamine-BHT, unsaturated ethenolamine-BHT, ethenolamine-BHPs gradually increasing in absolute abundance over the transition phase (i.e., aminotriol and N-acyl-aminotriols C<sub>12:0</sub>, C<sub>14:0</sub>, C<sub>15:0</sub>, C<sub>16:0</sub> and C<sub>18:0</sub>) (Fig. S5). The continuous presence of aminotetrol during this period is likely associated with methane-oxidizing bacteria as the pelagic redoxcline was established. In the Black Sea, evidence for aerobic methanotrophs during the transition phase is supported by the presence of ethenolamine-BHpentol, which has so far only been detected near a terrestrial methane seep and in lakes with low oxygen conditions in the bottom waters (Hopmans et al., 2021; Richter et al., 2023). Similarly, in the Baltic Sea, the transition from the Ancylus Lake phase to the Littorina Sea was marked with a peak in aminotriol and aminotetrol, attributed to methane-oxidizing bacteria (Blumenberg et al., 2013). The presence of MOB-associated BHPs would suggest that aerobic methane-oxidation occurred during this phase.

During the early marine phase, ethenolamine-BHpentol, ethenolamine-BHhexol, propenolamine-BHT, aminotetrol, and aminopentol are all present in higher relative abundance. While propenolamine-BHT has been observed near a terrestrial methane seep, as well as in lakes and in the sediments of coastal lagoons, its microbial source remains uncertain (Hopmans et al., 2021; Richter et al., 2023). Methanotrophic bacteria, known to thrive in the oxic-anoxic transition zone in the modern Black Sea water column (Kuypers et al., 2003; Durisch-Kaiser et al., 2005; Schubert et al. 2006), are known sources of aminotetrol and aminopentol (Blumenberg et al., 2007; Wakeham et al., 2007), however, the distributions and concentrations of BHPs detected in the modern-day Black Sea water column are not reflected in the geological record as there is no effective transport mode from the oxic-anoxic transition zone to the sediment (Wakeham et al., 2007; Blumenberg et al., 2009a; Kusch et al., 2022). However, the high absolute abundance of isorenieratene throughout the early marine phase indicates that euxinic conditions reached the photic zone during this period (Cutmore et al., 2025), pointing to a shallower oxic-anoxic interface compared to the later marine phase. The large accumulation of particles just below the photic zone at the oxycline likely led to a heightened transport of particulate matter, including BHPs, from the oxic-anoxic interface to the sediments (e.g., Repeta, 1993; Sinninghe Damsté et al., 1993; Coolen et al., 2008; Blumenberg et al., 2009a). Between 5 and 3.9 ka, SRB likely occurred in the deeper waters and sediments of the Black Sea and are another potential source for aminotetrol and aminopentol. This signal is unlikely to have been reflected in the sedimentary record as there is no effective transport of lipids from the deeper waters to the sediments (Schouten et al., 2001; Wakeham et al., 2003), and BHP-producing species of SRB were not detected in genetic analyses of a Black Sea sediment core (Blumenberg et al., 2006; 2009b; 2012; Leloup et al., 2006). Thus, unless there are previously unknown BHP-producing SRBs in the Black Sea sediments, SRBs are likely only a minor source of BHPs during the marine phase (Blumenberg et al., 2009a). N-acyl-aminotriols (i.e.,  $C_{12:0}$ ,  $C_{14:0}$ ,  $C_{15:0}$ ,  $C_{16:0}$ ,  $C_{17:0}$ , and  $C_{18:0}$ ) are also present in high absolute abundance during the early marine phase compared to the rest of the core and peak at 6.1 ka. There is no known distinct source for N-acyl-aminotriols; however, they have been detected in a methanotroph culture as well as various environmental settings (e.g., Hopmans et al., 2021; Kusch et al., 2022; Richter et al., 2023). The high abundance of these long-chain compounds might be associated with heightened transport from the oxycline to the sediments and increased preservation during this time (Blumenberg et al., 2009a; Cutmore et al., 2025). Thus, methane-oxidizing bacteria were likely active in the early marine phase near the oxycline of the Black Sea, as reflected in the sediment record.

High-rates of methane-oxidation in the modern-day Black Sea are mediated by anaerobic methanotrophs in the euxinic water column, with lower rates of anaerobic methane oxidation occurring in the sediments (Reeburgh et al., 1991; Lin et al., 2006; Schubert et al., 2006). MOB are present at the oxic-anoxic transition zone of the modern-day Black Sea water column; however, their associated rates of aerobic methane oxidation are several orders of magnitude lower than the anaerobic oxidation of methane (AOM) occurring in the anoxic water column (Reeburgh et al., 1991; Schubert et al., 2006). Consequently, the limited transport of suspended particulate matter from the chemocline and euxinic bottom waters to the sediments, leads to an underrepresentation of these lipids in the

geological record and restricts any lipid-based interpretations of methane-oxidizing microbial communities or associated SRB in the Black Sea water column during the late marine phase (Schouten et al. 2001; Wakeham et al., 2003; Schubert et al., 2006). Studies of lipid biomarkers associated with anaerobic methanotrophs (e.g., compound-specific carbon isotope compositions of archaeal lipids) confirm that there was minimal overprinting from active methane-oxidizing archaea in the sediments and fossil lipids derived from anaerobic methanotrophs were only a minor portion of the archaeal lipid assemblage during the marine phase (Zhu et al., 2024). In summary, MOB-associated BHPs in our record indicate that aerobic methane-oxidation occurred throughout the Black Sea record. However, archaeal lipids and modern studies demonstrate that anaerobic methane-oxidation became a more significant "methane sink" in the Black Sea water column during the marine phase relative to the lacustrine phase. Additional studies are needed to confirm this hypothesis by quantifying the contributions of aerobic versus anaerobic methanotrophs and identifying SRB during the lacustrine and transition phase.

#### 3.6. BHPs as indicators for surface salinity

Methoxy-BHT has previously been detected in marine environments and at present, has not been detected in lake samples (Richter et al., 2023). In the Black Sea record, we detect methoxy-BHT intermittently during the oxic lacustrine phase, possibly due to the slightly brackish conditions that existed during this period (Huang et al., 2021). After the IMI, the absolute abundance of this BHP increases, possibly due to the influx of saline water. The highest abundance of methoxy-BHT occurs between 6.1 and 4.2 ka, when surface water salinities reach a peak (Filipova-Marinova et al., 2013; Huang et al., 2021). The reduction in methoxy-BHT between 3.8 and 1.5 ka coincides with the non-detection of isorenieratene (Cutmore et al., 2025) which likely resulted from the erosion of the chemocline (Sinninghe Damsté et al., 1993) due to higher freshwater input, with a short reoccurrence of freshwater/brackish species occurring at this time (Filipova-Marinova et al., 2013; Huang et al., 2021). Consequently, the lower salinity of the surface waters may have also led to the decline in methoxy-BHT. Methoxy-BHT and methoxy-ethenolamine-BHT show similar patterns throughout this period (Fig. 6), both peaking between 6.1 and 4.2 ka, declining between 3.8 and 1.5 ka and increasing towards the end of the record. To our knowledge, methoxy-ethenolamine-BHT has not previously been linked to salinity, but its close correspondence with methoxy-BHT during this period indicates that both BHPs may be a specific adaptation of BHP compositions to saline environments or are linked to specific microbes adapted to brackish to marine systems.

#### 4. Conclusions

The Black Sea record shows substantial shifts in BHP absolute abundance and composition over the last 20 ka, which correspond to major environmental transitions. During the lacustrine phase (19.5 - 9.6 ka), high Nu-BHP absolute abundance and diversity likely reflects increased terrigenous input during the Last Deglaciation and/or in situ production in the water column. The transition phase (9.6 - 7.2 ka) is marked by a decrease in many BHPs, such as Nu-BHPs, and an increase in methoxylated-BHPs and more ubiquitous BHPs (e.g., aminotriol), likely linked to

changing oxygenation and salinity of the water column. The early euxinic marine phase (7.2 ka – 5 ka) is characterized by higher aminotetrol and aminopentol relative to the late marine phase, likely transported with sinking particulate matter to the sediment from a shallow oxycline. The late marine phase (5 ka – present) is marked by a decline in these BHPs, likely due to a deepening of the oxycline. BHP distribution changes throughout the record are attributed either to microbial adaptations to shifts in the oxygen levels or hydrology of the basin, or are associated with specific groups of bacteria suited to these conditions. For instance, BHT-CE was closely associated with crenarchaeol throughout the record, indicating that these compounds are either produced in a similar ecological niche or by related microbes. Aminotetrol, aminopentol, ethenolamine-BHpentol and -BHhexol were detected in the Black Sea record as likely biomarkers for methane-oxidizing bacteria, while methoxylated-BHPs, detected only during the transition and marine phases, may be associated with higher salinities or marine environments. The changes in the Nu-BHP distribution likely reflects changing terrestrial input over this period, with the exception of some Nu-BHPs (e.g., 2Me-adenosylhopane<sub>HG-diMe</sub> and Me-adenosylhopane<sub>HG-diMe</sub>) that are likely produced in the water column. BHP degradation products were detected as a minor component of the overall BHP distribution, suggesting good BHP preservation throughout the record. This new BHP record provides valuable insights into ecological changes as well as changes in the nitrogen and methane cycle in the Black Sea over the last 20 ka.

#### Data Availability

All data generated for this study are archived and publicly available via the Mendeley Data repository online at http://doi.org/10.17632/m6v5mj5gtp.2 (Cutmore and Richter et al., 2025).

#### **Author Contribution**

AC- Conceptualization, Formal analysis, Investigation, Data Curation, Visualization, Writing - Original Draft, Writing - Review and Editing. NR- Conceptualization, Formal analysis, Investigation, Data Curation, Visualization, Writing - Original Draft, Writing - Review and Editing. DR- Supervision, Writing - Review and Editing. NB- Supervision, Writing - Review and Editing. SS- Supervision, Funding acquisition, Writing - Review and Editing.

#### **Competing Interests**

The authors declare that they have no conflict of interest.

#### Acknowledgements

We thank the Chief Scientist Prof. Laura Villanueva and Dr. Rick Hennekam, as well as the captain and crew of the R/V Pelagia for the collection of core 64PE418. For laboratory support we thank Anchelique Mets, Denise Dorhout and Monique Verweij. Thank you to Diana Sahonero Canavesi for helpful discussions and Bingjie Yang for providing

helpful insight into the BIT index of this core. This study was funded by the Netherlands Earth System Science Centre (024.002.001) from the Dutch Ministry of Education, Culture and Science (OCW) and the Swiss National Science Foundation (SNSF) Ambizione Fellowship (PZ00P2\_216050).

493

495

490

491

492

# 494

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

# **Figures**

**Figure 1: a.** Site of Black Sea core 64PE418 taken at a water depth of 1970 mbsl (Adapted from: Giorgi Balakhadze, English Wikipedia, 2016); and a schematic diagram of the Black Sea basin during: **b.** the marine euxinic phase; **c.** the transition phase; **d.** the oxic lacustrine phase.

#### a. General bacteriohopanepolyol (BHP) structure

#### b. BHP core structure modifications:

26- or 36-methyl 
$$\Delta^6$$
 or  $\Delta^{11}$ 

#### C. BHP side structure modifications:

#### Bacteriohopanetetrol-cyclitol ether

Anhydrobacteriohopanepolyols

R<sub>4</sub>= H: Anhydro-BHT R = OH: Anhydro-BHpentol Tetrakishomohopane-32,33,34-triol

Saturated: BHtriol Unsaturated: Unsat. BHtriol

#### Ethenolamine- and propenolamine-bacteriohopanepolyols

$$\begin{array}{c} R_1 \text{ and } R_2 = H, \, R_3 = \text{OH}, \, R_4 = \text{NH}_2\text{: Ethenolamine-BHT} \\ R_1 \text{ and } R_2 = H, \, R_3 = \text{OCH}_3, \, R_4 = \text{NH}_2\text{: Methoxy-ethenolamine-BHT} \\ R_1 = H, \, R_2 \text{ and } R_3 = \text{OH}, \, R_4 = \text{NH}_2\text{: Ethenolamine-BHpentol} \\ R_1, \, R_2 \text{ and } R_3 = \text{OH}, \, R_4 = \text{NH}_2\text{: Ethenolamine-BHhexol} \\ R_1 = H, \, R_2 \text{ and } R_3 = \text{OH} \text{ and } R_4 = \text{CH}_2\text{NH}_2\text{: Propenolamine-BHT} \\ \end{array}$$

#### Aminobacteriohopanepolyols

 $R_a$  and  $R_a = H$ :

35-aminobacteriohopane-32,33,34-triol (aminotriol)

 $R_{a} = H$  and  $R_{a} = OH$ : 35-aminobacteriohopane-31,32,33,34-tetrol (aminotetrol)

 $R_a$  and  $R_a = OH$ :

35-aminobacteriohopane-30,31,32,33,34-tetrol (aminopentol)

### N-acylated-aminobacteriohopanetriols

#### N-formylated-35-aminobacteriohopanetriol

Oxazinone-aminobacteriohopanetriol

Dioxanone-methylaminobacteriohopanetriol

### d. Other compounds:

Crenarchaeol

Isorenieratene

1028

Figure 2: (A) A general bacteriohopanepolyol (BHP) structure with the core hopane and extended side chain (example of BHT). (B) Common modifications to the core BHP structure, including additional methyl groups and unsaturations on the core structure. (C) Side chain configurations discussed in this study. Note: for methoxy-BHT and methoxy-ethenolamine-BHT, the placement of the methoxy group is arbitrary. (D) Structures of other compounds discussed in this study: crenarchaeol and isorenieratene.

**Figure 3:** Results from a principal component analysis of the bacteriohopanepolyol (BHP) abundance with depth. The individuals plot shows the sample depths with the ellipses highlighting major groupings. The biplot shows the sample depth (colored circles) and the contributions of each BHP (where BHT = bacteriohopanetetrol, unsat. = unsaturated, Me = methyl, and HG = headgroup). Note: BHPs with low contributions (

**Figure 4: a)** Relative abundance of peak areas (%) of all BHPs identified in core 64PE418 over the last 19.5 ka. Changes in abundance of each BHP (peak area per g TOC) over time is shown in the supplementary material. 'Early' and 'Late' Me-adenosylhopane<sub>HG-diMe</sub> represents two different isomers'; **b)** Total BHPs represents the sum of all BHPs (in peak area, pA) identified in the sediment samples and normalised to grams of dry sediment (pA g<sup>-1</sup>).

**Figure 5:** Changes in the peak area per g TOC of Nu-BHPs in Black Sea core 64PE418: **a)** over the past 19.5 ka; highlighting changes during: **b)** the oxic lacustrine phase; **c)** the transition phase; and **d)** the euxinic marine phase. Note the different scales on the y-axes. 'Early' and 'Late' Me-adenosylhopane<sub>HG-diMe</sub> represents two different isomers.

Figure 6: Changes in the peak area per g TOC of a number of potential diagnostic BHPs.