# Peer review of "Bacteriohopanepolyols track past environmental transitions in the Black Sea"

_EGUsphere, 2025_

## Author Response (AR1)

Dear Dr. Sebastian Naeher,

Thank you for your helpful and constructive feedback. Please find a list of all the relevant changes made to the manuscript in response to the reviewer comments. These changes are also made clear in the tracked-changes document.

Thank you for your consideration of our research.

Yours sincerely, Dr. Anna Cutmore, on behalf of co-authors

**Editor comments,**

• I wondered about terrestrial inputs from the Danube River. You discuss terrestrial inputs and sources and large terrestrial contributions would be expected from the NW shelf of the Black Sea, so could the terrestrial OM contributions largely originate from material contributed by the Danube River? Maybe an additional note about more specific sources of terrestrial material could be added? But these are very minor points that you could consider to include or not based on your preference

Thank you for this comment. We have added information about the major sources of the terrestrial organic matter to the Black Sea, "primarily supplied by three major rivers: the Danube, the Dnieper, and the Don"

**Reviewer 1,**

**General points:**

• First, the determination of BHP concentrations is complicated by the certainly very different responses of individual BHP during ionisation and decay during mass spectrometric analysis. The authors correctly describe this. However, they withdraw from this and write that they cannot make any (semi-)quantitative statements. In practice, however, they do it indirectly themselves by adding up the peak responses (in relation to TOC) in Figure 3, for example. If no quantitative comparisons are permitted, a presentation that totals up to 100 % is also out of the question. I think it should be possible to use this data and also make a semi-quantitative statement on the relative concentration of at least the majority of BHPs versus time/depth (with mentioning of the restriction). I would find such a curve helpful as insert to Figure 3 (as a or b). Of course, it would also be interesting to see whether this is consistent with other studies that have analysed (fewer) BHPs, but where there is a large overlap in the BHP biomarkers used and most common?

Thank you for this comment. In our manuscript we do make "semi-quantitative" interpretations, as that is one of the few ways that we can discuss this dataset. This type of data is very similar to DNA datasets, which are also often discussed in the context of relative abundances even though this does not directly reflect the actual abundances of species present in a dataset. We think it is important to highlight the limitations of the dataset, even though we do rely on "semi-quantitative" interpretations. We have rephrased the sentences discussing these limitations in the methods section to make this clearer. Further, we have included a second subplot in Figure 3 to show "total BHP concentrations." However, we would like to refrain from making comparisons with previous studies about total BHP concentrations as these studies used either GC-MS or HPLC-APCI-MS quantifications whereas our study uses an UHPLC-ESI-HRMS system. The ionisation efficiency

in all three systems differs, and unfortunately, we do not have any standards that would allow us to make this comparison.

• Second, there seem to be only a few errors or ambiguities here, but the authors should check this again very carefully. For example, the term "anhydrous-" BHT is sometimes used and sometimes "anhydro-" BHT.

We changed "anydrous-BHT" to "anhydro-BHT" throughout the text

• Furthermore, it remains unclear to me what "BHT" means in Figure 3, for example? In S4, the course of BHT-22S is shown. Is the former a sum of the different BHT isomers? Please do not take this as a request to show all isomers. This would complicate the manuscript even more. However, how certain is the structural elucidation of "BHT-22S" really? Are there, for example, co-elutions with BHP extracts from reference organisms? Me-adenosylhopaneHG-diMe exists twice in Figure 4, for example (in addition to 2-Me and 3-Me). Are these isomers? Which ones are meant in the text when the compound is mentioned? Please double-check everything again. Further, a correct "...may be..." to a structure proposal in a cited study ("BHT-22S") does not seem to be discussed here and the structure or interpretation of the structure is simply adopted (text and Fig. S4). Of course, it is not possible to simply repeat the very extensive structural elucidations that colleagues from Strasbourg in particular carried out on individual BHPs in the 1980s and 1990s. Nevertheless, the limitations of structure elucidation with MS-(MS) should not be forgotten and uncertainties should be described to remind readers to this limitation.

In BHT in Figure 3 is meant to represent regular BHT (BHT-34S) and BHT-x is shown separately in the figure. We have changed the label in the figure 3 caption from BHT to BHT-34S.

We agree with the reviewer that using a MS-(MS) based system we are limited in our structural identifications, and as we state in the methods sections these are "tentative" identifications. Our structural identifications are based on retention times, exact masses, and fragmentation spectra, as well as comparisons with previous studies. In the case of the BHT isomers, the identification of these compounds and distinguishing the various isomers on a UHPLC system was extensively discussed in Schwartz-Narbonne et al. (2020) and Hopmans et al. (2021). Identification of the different isomers was determined based on retention times as described in Cutmore et al. (2025), which was found to be reliable when using a UHPLC system (Schwartz-Narbonne et al., 2020; Hopmans et al., 2021). We have added a short sentence to the methods section that clarifies these limitations and that we relied on previous studies for our structural elucidations.

Thank you for catching this mistake with the labelling of Me-adenosylhopaneHG-diMe. The two Me-adenosylhopaneHG-diMes in Fig. 4 refer to two different isomers with unknown Me positions, therefore, we have changed this to "early" and "late" isomers in Figures 3, 4, and S8 as well as in Table S1.

• Third, as a reader, I miss take-home messages in some places in the sub-chapters. In some cases, relatively long explanations with pros/cons/exceptions are described (please utilise the potential for shortening here). However, a quintessence is sometimes missing. One example is the long block of text on "BHPs associated with the N-cycle". As a reader, I would expect a clear categorisation of whether the data fully correspond to the interpretation in Cutmore et al. (2025), whether they are helpful and supportive for the other paper or whether they contradict aspects of it. What is new and different in this manuscript compared to Cutmore et al 2025?

Thank you for this comment. We have altered this chapter and clearly referenced the CoP paper rather than outlining the findings in detail. We have also clearly shown what is new research/conclusions and outlined how the new findings support the interpretations of Cutmore et al., 2025. Finally, we have ensured there is a concluding sentence.

Finally, in general, the authors demonstrate a great deal of expertise and generally cite existing studies that
have worked on similar issues (including with BHP, albeit with fewer structures) in a good way.
Nevertheless, at one point or another I would expect categorisations as to where there is clear support
from the new data and where there is not.

Thank you for this comment. We have made sure we have concluding sentences and made it clear throughout the manuscript where our study is supporting existing information, and where we are putting forward new theories.

• Line 44: I think early studies by Helen Talbot should be mentioned in these citations of important papers (from 2003 or similar). Her work was methodologically groundbreaking, but also contributed to the understanding of source organisms and biogeochemical processes.

We're grateful for your recommendation. We have mentioned these key studies by Talbot et al. (Talbot et al., 2003, Organic Geochemistry; Talbot & Farimond, 2007, Organic Geochemistry)

• Line 88: Please replace "compounds" with "BHPs".

Thank you for this. We have replaced this with BHPs.

• Line 97: mbsl must be explained.

Thank you for picking this up. We have made sure this acronym is explained as metres below sea level

• Line 130: See general note. This is understandable, but if you take a very strict view, then BHPs can also not be shown in a comparative graph (e.g. Fig. 3). However, this greatly reduces the informative value, as only the progressions of individual connections can then be considered. I suggest looking for a way to show at least the total BHPs over time in a semi-quantitative way.

As discussed above, we have amended the text to highlight that although our data is only "semi-quantitative", it is still valuable to show major changes in BHP abundances as it demonstrates changes in BHP distributions during the key periods of the Black Sea. We have also added a subplot to Figure 3 to show changes in total response units of all BHPs per gram TOC. Comparisons of individual BHP abundances are shown in the supplementary which we will have made sure are clearly referred to in the text when necessary.

• Line 173: See above. Is it really BHT-22S? The illustration only says BHT. It seems to me that the latter is better documented.

Thank you for noticing this. It is regular BHT (BHT-34S) that we have identified based on the mass spectral information. Consequently, we have changed 'BHT' to 'BHT-34S' in Fig. 3 to improve clarity.

• Line 197: See above. This statement refers to "absolute abundances of BHP", but this is not shown at all. Please find a way and discuss the result comparatively in this chapter.

We have rephrased this and referred to the supplementary graphs where this is shown.

• Line 211: Please change to "...suggesting rapid early diagenetic modifications of BHPs."

We have rephrased the sentence accordingly.

Line 237: "later"

Thank you for picking this up, we have made this suggested change.

Line 279ff: "As future studies discover more..." sounds odd. Please check and consider rewriting.

We have rephrased the sentence as follows: "As ongoing research continues to explore the sources of Nu-BHPs"

• Lines 334ff and 380ff: I think it is very unlikely that a relevant amount of BHP comes from sulphate-reducing bacteria. For the lacustrine phase, this can easily be explained by the general absence of relevant sulphate in the water. It would be good if the authors could find a way a shorter way to have this component-specific discussion of why the one is somewhat unlikely, though not impossible, and so on. Similar to the authors, I think that neither the transport of biomass from the sulphidic zone nor relevant biomass in the sediment plays a major role. The text and a (shorter) length should express that.

Thank you for this feedback, we have shortened this section.

• Line 356: Aminotriol seems so unspecific to me that it could be produced by all kinds of bacteria.

We agree that aminotriol is found in too many bacteria to make this conclusion, so have removed this from the manuscript.

• Lines 359-366: It is not entirely clear what is meant here. Is there now a peak for methanotrophic bacteria in the transition phase? Does this show aminotetrol and pentol? The study cannot contribute anything to the anaerobic oxidation of methane and the topic should therefore only be mentioned briefly.

We have amended our phrasing to more clearly state that we observe MOB-related BHPs throughout the transition phase (i.e., aminotetrol and ethenolamine-BHpentol). Although there is only a small peak in these BHPs near the beginning of the transition phase, they are present throughout this time period. We have removed the sentence on anaerobic methane oxidation.

• Line 366: The data does not show a "distinct increase" for me. Please describe a little better. Figure 5 shows a peak before the transition phase and, in comparison, slightly increased values in marine phase II.

Thank you for pointing this out. We agree this is confusing, as ethenolamine-BHpentol, ethenolamine-BHhexol and propenolamine-BHT all show an increase, while the aminotetrol and aminopentol show a more complex change. We have rephrased this sentence to make it less confusing and more accurate.

• Line 441: This statement is difficult to understand based on the data. Aminotetrol and -pentol appear to fluctuate strongly (Figure 5).

We have rephrased this section to make it clear that we are talking about absolute abundance being higher during the early marine phase compared to the later marine phase.

Figure 1: Please write the approximate ages (in ka) next to the models for the three phases.

Thank you for this suggestion, we have included the ages in this figure.

• Figure 3: Many BHP structures are colour-coded in the figure. It is sometimes difficult to make an assignment and it is even impossible without colour (black/white). Perhaps the authors could write the abbreviations of the BHP on the main horizontal bars to at least make these clearly comprehensible?

We appreciate the suggestion. To maintain the visual clarity of Figure 3 and highlight broad temporal patterns, we prefer not to add text to the bars, as it may make the figure overly cluttered. The detailed information on individual BHPs is provided in the supplementary material and has been referenced more clearly in the main text. We have also added a supplementary figure showing diagenetic products to help visualize changes in specific BHPs more clearly. Based on the reviewer's previous comments we have added a subplot to this figure to show "total BHP abundance" and a supplemental spreadsheet containing all the data so readers can confirm trends shown in the figures.

**Reviewer 2,**

• The role of degradation products of originally produced BHPs is also discussed in one chapter (4.2) in this paper, however, it only includes very few hopanoids, like anhydro BHT, anhydro BHpentol, as well as some BHtriols. These compounds are only very early degradation products, and only allow a small portion of the degradation story of hopanoids. Since the authors would like to report about these signatures, I suggest to modify the presentation of these compounds in Figure 3. For me it was not easy to see the diagenetic products in this figure. I suggest to make a new group, or to show the diagenetic BHPs with hatches, points or whatever, additionally to the colour code already available. So everyone can see at first sight which BHPs are diagenetic BHPs. More suggestions can be found in the general comments.

We're grateful for your recommendation. We have added a new supplementary figure to show changes in the diagenetic products to help visualize changes in these BHPs more clearly and refer to this figure in chapter 4.2.

• Line 54: ...oxidizing...

Thank you for noticing this, we have made the change.

• Line 65: This is true, but is limited by the stability of the BHPs, which is already demonstrated by the presence of early degradation products. The authors, however, do not mention the possibility, that at least a certain percentage of the BHPs is not in the easy extractable fraction, but may be preserved as macromolecules, especially under euxinic conditions as organic sulfur compounds. This is not the topic of this paper, but should at least be mentioned, that only very early degradation products are monitored, so it is rather an incomplete observation and discussion of degradation products of BHPs. This must be clarified in chapter 4.2. Secondly, as already mentioned by the authors, the record of many of the measured BHPs is very limited over time, and can only be found in sediments a little older than 1 million years.

We have added a sentence to section 4.2 to highlight that this study only focuses on early "polyol" degradation products of BHPs and that potentially S-bound BHPs were not extracted. We have also mentioned that this paper does not discuss the additional degradation processes of BHPs towards more stable hopanoid products (as these were not analysed using our method). As to the reviewer's second point, we agree that BHPs have so far only been identified in continuous paleo-records that span the last 1 million years (e.g., Talbot et al. 2014) and are likely more degraded on longer timescales. However, as our study only looks at BHPs over the last 20ka we do not think this is a major limitation in our study. We have added the word "recent" before "geological record" in the text.

• Line 79: replace 'enhanced' by 'increased'. This should be done all through the manuscript.

We have made the change throughout the manuscript.

General comment Chapter 4.2: As mentioned in the general comments above the diagenetic products here
are only covering a very small portion of the diagenetic pathway of BHPs. Blumenberg et al. (2009) tried to
unravel this problem already in their manuscript and their findings should be mentioned more clearly,
especially when discussing the degradation products here.

We have added a clearer discussion on the diagenetic products as described by Blumenberg et al. (2009).

• Line 217: I am not sure if everyone is aware of the BHtriols, so the papers first reporting about them must be mentioned, as well as the environments they were found (e.g. Watson and Farrimond, 2000 or papers from Lago di Cadagno). As much as I can remember, they are especially found in lacustrine samples, which seems to be true also for the Black Sea samples.

Thank you for the suggestion. We have added these key references and mention the previous environments in which BHtriols have been found.

• The potential sources of the degradation products anhydro BHT have been at least partially identified in culture and P/T experiments, as cited by the authors, but to my knowledge BHtriol and other related compounds were never produced in any experiments, so not a lot is known about them. Maybe I overlooked this in the literature, but if there is anything known it should be discussed as well. Most likely, both anhydroBHT and BHtriol seem to be derived from BHPs with four functional groups, whether this was a compound like adenosylhopane or just BHT or methoxy BHT, is unclear. BHPs with 5 or 6 functional groups are rather unlikely, since they had two additional hydroxy groups at carbons 31 and 30. In Watson and Farrimond (2000) they also showed diols and triols with hydroxy groups at positions 30 and 31. If there are BHPs with functional groups at C-30 and C-31, I would expect to find such degraded BHPs as well. Is this the case? Were any of these 'degraded' hopanols introduced by Watson and Farrimond (2000) found?

To our knowledge, Rodier et al. (1999) and Watson & Farrimond (2000) are the only previous studies on these compounds. As suggested by the reviewer, we searched for the additional diols and triols identified in Watson & Farrimond (2000); we were only able to tentatively identify one of the triols with 32 carbons  $(C_{32}H_{56}O_3)$  in the lake phase. It is possible that other diols or triols were present during the lake phase, however, the fragmentation patterns were not clear enough for us to assign these compounds. We have mentioned this in the text.

• Figure 3: The figure is very nice and it's easy to follow the various colour codes. However, when reading the text I found it rather difficult to identify the diagenetic BHPs. Maybe it is possible to make either a separate group of columns only for these BHPs, or alternatively the degraded BHPs could be marked by hatching, so everyone can see immediately what are diagenetic BHPs. See my general comments.

We thank the reviewer for their suggestion, we have added an additional figure that shows the diagenetic BHPs to the supplementary material.

• Line 239: I am confused by this reference. The element and GDGT distributions are both from Hopmans et al. (2004)? Which data are from Yang and personal communication? Element data are also provided in Cutmore et al. (2025). Please clarify. Further, the authors should refer here to their figure 5, where crenarchaeol is displayed. I suggest to include the BIT curve next to the crenarchaeol as well, so anyone can follow better what the authors are talking about.

Thank you for identifying this. We have altered this sentence to improve clarity, to ensure the reader is aware that the elemental and crenarchaeol records are from Cutmore et al., 2025 and the BIT index is from Yang

(personal communication). The BIT curve is a key part of an upcoming manuscript by Yang et al., so is unable to be shown in this study.

• Line 252 ff.: Is there anything known about potential producers of 2Me-adenosylhopaneHG-diMe? What is the basis for the interpretation, that this compound is from bacteria thriving in the water column? Its interpretation as indicator of warming is rather speculative. I am not fully convinced and suggest to tone this down.

There are currently no known producers for 2Me-adenosylhopaneHG-diMe, although this is part of ongoing work at NIOZ. The hypothesised production of 2Me-adenosylhopaneHG-diMe in the water column was based on previous observations in a lake, where 2Me-adenosylhopaneHG-diMe increased with depth (Richter et al., 2023). Further, in the Black Sea record, 2Me-adenosylhopaneHG-diMe abundances vary independently of other nucleoside BHP distributions before the transition phase (Fig. S8 & S9), and also differs from other proxies that are indicative of soil inputs during this time period (Fig. 5). Thus, we speculate that 2Me-adenosylhopaneHG-diMe is derived from an alternative source during this time period. We have added a short sentence to clarify this point in the manuscript and we have removed the section that suggests it is associated with warming temperatures.

• Lines 261, 270: It is okay to take data from another paper (Ti/Ca, K values), but they need to be cited, or shown in figures as well. Such information is important. I know it has been done somewhere else, but need to be provided whenever needed throughout the text.

Thank you for pointing this out. We have added the citation (Cutmore et al., 2025) every time these records are mentioned in the text.

• Line 287: This sentence is confusing. It reads like the Anammox are expanding from the sediment to the water column. Re-write the sentence.

To make sure this is clearer, we have rephrased this sentence as follows: "which enabled anammox bacteria to inhabit both the anoxic sediments and overlying water column"

• Line 295 ff.: There is a lot of speculation about the source of BHT-CE. Nitrososphaerota are suggested as producers. Is it known that these bacteria can produce BHT-CE? Refer to a paper, or explain that this is a speculation based on whatever. BHT-CE and potential sources seem to be puzzling, also in other studies. It is intriguing, that both crenarchaeol and BHT-CE really seem to correlate well over the entire core, which is surprising, because Thaumarchaeota and the potential BHT-CE producers are possibly not thriving in exactly the same environment. BHT-CE are either known from soil, peats etc. produced by unknown anaerobic bacteria, or methylotrophic bacteria.

Thank you for these comments. We have made it clearer in the manuscript that Nitrososphaerota are archaea and therefore not suggested as producers of BHT-CE. The suggestion is that the dominant bacterial producer of BHT-CE was coupled either to the archaea Nitrososphaerota or to the first step in the nitrification process. We have made sure this explanation is clearer. We are also extremely surprised and interested by this coupling and it certainly would be a great area to explore in future studies.

• Line 312 ff.: The old reports of 2-methyl hopanoids produced by marine cyanobacteria are outdated, and have been disproved, rather other producers are more likely. Finally, the authors decide to assign them potentially to heterocyst cyanobacteria? Do HGs produce 2-methyl hopanoids at all? This suggestion must be confirmed by any data reported in the past. Numerous studies analysed a great variety of cyanbacteria starting from Talbot et al., 2008, and it was the major topic of the paper by Naafs et al., 2022 as well, to verify sources of 2-methyl hopanes in various environments. Findings presented in these papers need to be discussed and included also here.

Thank you for these suggestions. We have made it clearer in our manuscript that the reports of 2-methyl hopanoids produced by marine cyanobacteria were initial propositions by early studies, and that, since then, there has been development and new theories of who is producing these BHPs. Furthermore, we have added information about the heterocystous cyanobacteria that have been shown to produce 2-MeBHT (Nostoc muscorum, Calothrix sp. and Chlorogloeopsis fritschii) from these key studies.

• Chapter 4.4: It is great to discuss the sources in detail, but especially for the BHP inventory tentatively associated with the N-cycle, there is a lot of overlap with the paper published by the same authors last year in Climate of the Past. I am aware that this paper is supposed to cover all potential sources of BHPs and need to include also the BHPs associated with the nitrogen cycle, but I suggest to shorten this chapter and refer to the Climate of the past paper.

We're grateful for your recommendation. We have shortened this chapter to ensure there is no overlap, and referenced the CoP study.

• Line 344/345: Be more precise with the MOB. MOB can thrive in various environments but in the setting described here they are especially abundant at the chemocline, better oxyclines. If MOB are prominent in the sediments under fully oxic conditions, this needs to be further discussed. There are reports of MOB in methane seeps, especially in seep carbonates formed in anoxic sedimentary conditions, but microaerophilic niches where MOBs can thrive (see Cordova-Gonzalez et al., 2020 and references therein). It must be further discussed, if there is additional evidence from other lipids, that methane was oxidized in the lacustrine phase. If you discuss sedimentary sources of MOB BHPs, these occurrences must be (at least shortly) further discussed.

We agree with the reviewer that MOBs in lakes are especially abundant near the oxycline, however, MOBs are known to thrive in fully mixed lakes (i.e., when the water column is oxic) at the sediment-water interface (see Hanson & Hanson, 1996 and references therein), but have also recently been detected in anoxic lake sediments (e.g., Martinez-Cruz et al., 2017). Thus, a sedimentary source of BHPs during the lake phase cannot be excluded, although we agree with the reviewer that this is likely a minor source of BHPs relative to the water column. Further work on BHPs in lakes is needed to distinguish these sources, and as far as we know, there are currently no additional studies on this topic. We have added a few references from modern lake studies to highlight sedimentary BHPs from MOB as a potential source.

• Line 363: The citation of Zhu et al., 2024 is tricky. The report of 13C-depleted carbon isotopes is true only for archaeols, but not for the GDGTs Caldarchaeol and Crenarchaeol, respectively acyclic biphytane and tricyclic biphytane. These two GDGTs show isotope values characteristic for marine planktic Thaumarchaeota. Isotopes of monocyclic and bicyclic biphytane, which are very abundant in ANMEs were not shown by Zhu et al., 2024. In deeper sediment, usually ANME-1 consortia are prevailing, and they produce especially

GDGT- 1 and GDGT-2, leaving behind isotopically depleted monocyclic and acyclic biphytanes after ether-cleavage. If ANMEs would have been important, they would have had some influence on the water column derived signature for the caldarchaeol, whereas the crenarchaeol should remain unchanged. This also questions the interpretation of the BHP signatures without isotopes, whether these are either signatures of bacteria from the chemocline or diagenetic sedimentary sources. One way to test the potential abundance of ANMEs in your sediments would be to calculate the methane index from the GDGTs (Zhang et al., 2011). Since Crenarchaeol has been measured, all other GDGTs were measured as well, I guess.

We have removed this sentence from the manuscript, as we agree with the reviewer that preservation may be an issue. The methane index in the Black Sea is complicated as any lipids produced in the bottom waters during the transition or marine phase do not necessarily get transported to the sediment. Thus, we have refrained from discussing this in the manuscript.

• Line 391: No, BHT-CE is definitely not only produced by SRB, the same is true for aminotetrol and aminotriol. Sure, Desulfovibrio can make these BHPs, but compared to contents in MOBs, these two are very minor BHPs, although the results from cultures are very limited. This should be mentioned here. The Talbot and Farrimond (2007) is a very good review, but more recent findings of BHT-CE and BHT-22S must be included and also discussed, such as Eickhoff et al. (2013), who report about BHT-CE in Geobacter, which also could be potential producers, or bacteria like M. oxyfera and related (e.g. Kool et al., 2012), but the latter were already mentioned above.

We're grateful for your recommendation. We have rephrased this section and added these references.

• Line 403: unclear wording. It reads like the rates of MOB in the anoxic water column are higher as in the oxycline, but here MOB vs. AOM is meant. Re-write this sentence.

Thank you for this comment, we have rephrased the sentence as follows: "MOB are present at the oxicanoxic transition zone of the modern-day Black Sea water column; however, their associated rates of aerobic methane oxidation are several orders of magnitude lower than the anaerobic oxidation of methane (AOM) occurring in the anoxic water column"

• Line 411, 412: Yes, this is true. They could not find any evidence for AOM, but honestly, they did not show any isotope data of the characteristic ANME biomarkers such as GDGT-2 (monocycic biphytane after ether cleavage) and archaeols, especially hydroxyarchaeols. See my comment for line 363. But in general I agree.

Thank you for the comment, we have removed the discussion at line 363 as detailed in our previous response but have left this in as a concluding discussion.

• Figure 3: It's in places confusing, a) when it comes to the discussion of the diagenetic BHPs (see my comments above), but especially for some of the less abundant BHPs, which are not so easy to be identified.

Thank you for the suggestion. We have added a supplementary figure to show changes in the diagenetic products to help visualize changes in these BHPs more clearly. Additionally, we have made it clearer throughout the text that the information on individual BHPs is provided in the supplementary material. Due to the large number of graphs, it is not possible to add these to the main manuscript.

• Figure 4: The numeration must be modified. The figure in the upper left corner needs a letter, too, even though it is just showing the same information as shown in the three graphs on the right hand side. Otherwise, the caption is incomplete.

Thank you, we have edited the graph and the caption accordingly.

• Figure 5 (new figure 6): This is a good way to show the potential candidates for specific groups of bacteria. I suggest to show the structures of the displayed BHPs and also the creanarchaeol, so everyone can see the major differences of the various BHPs and their potential producers. The readership of biogeosciences may not be aware of the various BHPs and other molecules and would be a valuable addition.

We're grateful for your recommendation. We have moved Supplementary Figure 1 (illustrating the BHP structures) to the figures of the main manuscript. To this diagram we have also add the structures of Crenarchaeol and isorenieratene.